



# An LES Exploration of the Assumptions used in Retrieving Entrainment from a Mixing Diagram Approach with Ground-Based Remote Sensors

Tessa E. Rosenberger[1], Thijs Heus[1], Girish N. Raghunathan[1], David D. Turner[2], Timothy J. Wagner[3], and Julia M. Simonson[2,4,5]

[1]Cleveland State University, Cleveland, OH, USA
[2]NOAA Global Systems Laboratory, Boulder, CO, USA
[3]University of Wisconsin - Madison, Madison, WI, USA
[4]Cooperative Institute for Research in the Environmental Sciences, University of Colorado, Boulder, CO, USA
[5]Developmental Testbed Center, Boulder, CO, USA

**Correspondence:** Tessa Rosenberger, t.e.rosenberger@csuohio.edu

**Abstract.** Entrainment is a crucial component of the atmospheric boundary layer (BL) moisture and heat budget. While usually thought of as only entrainment flux, entrainment within the mixed layer budget equation is really composed of two terms: the flux of a property across the boundary separating the BL from the free troposphere and the change in the concentration of a property as the depth of the BL changes. In a recent study, Wakefield et al. (2023) used ground-based remote sensing observations to estimate entrainment flux as the residual of a mixing diagram framework that was applied to the daytime convective boundary layer. This present work uses LES to examine how well this residual assumption for entrainment fluxes alone compares to the actual sum of those two entrainment terms derived from spatial averages of the LES output. We highlight the importance of the second entrainment term in closing the mixed layer budget and show that the residual assumption does not represent entrainment flux only but rather a total entrainment term when the boundary layer depth is changing.

## 1 Introduction

The atmospheric boundary layer (BL) is the section of the atmosphere that interacts directly with the surface and is responsible for the majority of our weather. Temperature and moisture changes within the BL can impact cloud formation (Ek and Mahrt (1994); Findell and Eltahir (2003); Ek and Holtslag (2004)), heat waves and droughts (Miralles et al. (2014); Miralles et al. (2019); Schumacher et al. (2019); Benson and Dirmeyer (2021)), and reintensification of tropical cyclones over land (Emanuel et al. (2008); Arndt et al. (2010); Andersen et al. (2013); Andersen and Shepherd (2013); Wakefield et al. (2021)). Therefore, more accurate representations of the evolution of the heat and moisture budgets in the BL is crucial for improving climate models, weather models, and ultimately forecasting extreme weather earlier.





At the top of a BL, free tropospheric air is incorporated down into the turbulent mixed layer through entrainment (Stull and Eloranta, 1984). van Heerwaarden et al. (2009) found that the entrainment of heat at the top of the BL directly increases

the depth of the BL, and dry-air entrainment enhances surface evaporation, which impacts cloud formation, exposing a need for entrainment to be accurately handled in models. While an important feature, entrainment is difficult to capture through observations due to its small scales of motion compared to the convective mixing within the BL (Cooper and Eichinger (1994); Nelson et al. (1989); Crum et al. (1987); Angevine et al. (1998)), position at the top of the BL - making it more difficult to measure with ground-based observations (Vilà-Guerau de Arellano, 2004), and the difficulty in computing horizontal averages

due to instrument spacing or differing terrain (Driedonks, 1982). Determining a more robust way to estimate entrainment fluxes from ground-based observations would lead to better understanding of the changes of temperature and moisture in the mixed layer.

One method for estimating entrainment was shown in Wakefield et al. (2023) (hereafter W23), using a mixing diagram framework inspired by Betts (1992). Betts developed the mixing diagram method for analyzing aircraft data in order to study

the evolution of the daytime mixed layer, using vector representations of heat and moisture budgets. This method was later applied to evaluate land-atmosphere couplings by Santanello et al. (2009). In a mixing diagram, the evolution of the heat and moisture budgets are plotted on the same figure as the vector components that make up that evolution, offering a visual representation of the contributions of various forcings to the changes within the mixed layer. Betts (1992) defined the mixed layer budget for an atmospheric property $\phi$ as:

$$\frac{\partial \overline{\phi}}{\partial t} = \frac{\partial \overline{\phi}_{ml}}{\partial t}\bigg|_{LS} + \frac{\overline{w'\phi'}}{z_i}\bigg|_{Sfc} - \frac{\overline{w'\phi'}}{z_i}\bigg|_{top} + (\frac{\partial z_i}{\partial t} - \overline{w_i})\frac{(\overline{\phi}_{top} - \langle\overline{\phi}\rangle)}{z_i}, \tag{1}$$

where overbars indicate horizontal averaging and angle brackets indicate averaging over the depth of the BL. The terms in this equation are:

$\frac{\partial \overline{\phi}}{\partial t}$ is the total change of the atmospheric property in the mixed layer over time

$\frac{\partial \overline{\phi}_{ml}}{\partial t}\bigg|_{LS}$ is the average large scale forcing across the BL (usually the advection term, but also radiative heating)

$\frac{\overline{w'\phi'}}{z_i}\bigg|_{SFC}$ is the atmospheric property flux at the surface over time

$\frac{\overline{w'\phi'}}{z_i}\bigg|_{top} + (\frac{\partial z_i}{\partial t} - \overline{w_i})\frac{(\overline{\phi}_{top} - \overline{\phi}_{ml})}{z_i}$ is the total entrainment. Within this total entrainment term, there is the entrainment flux (ENT1) term $\frac{\overline{w'\phi'}}{z_i}\bigg|_{top}$

and a second term that considers the difference between the magnitude of a property within the mixed layer and at the top of the BL $(\frac{\partial z_i}{\partial t} - \overline{w_i})\frac{(\overline{\phi}_{top} - \overline{\phi}_{ml})}{z_i}$, which we refer to as entrainment 2 (ENT2). ENT2 depends on the change in boundary layer



depth over time ($\frac{\partial z_i}{\partial t}$), the subsidence ($\overline{w_i}$), and the difference between the mean value of a property at the top of the BL and

the mean value within the mixed layer ($\overline{\phi}_{top} - \overline{\phi}_{ml}$). ENT1 is the flux of a property entering the boundary layer from the free

troposphere above, while ENT2 takes into consideration the change in BL properties due to the change in the BL depth over

time.

W23 showed that if the total evolution, surface fluxes, and average large-scale forcing are known, entrainment fluxes can be

estimated as the residual or closure term. In that study, the thermodynamic profiles were retrieved using the TROPoe retrieval

algorithm (Turner and Löhnert (2014); Turner and Blumberg (2019)) from radiance observations made by the Atmospheric

Emitted Radiance Interferometer (AERI; Knuteson et al. (2004)) at the central facility of the Atmospheric Radiation Mea-

surement (ARM) Southern Great Plains (SGP) site (Sisterson et al., 2016). Because of the vertical resolution of the profiling

instruments used in W23, they computed the mean of the mixed layer properties only from $0.1z_i - 0.5z_i$, where $z_i$ is the depth

of the BL determined from the TROPoe retrievals using a parcel method. Surface fluxes came from a combination of sur-

face flux measurements from the Eddy Correlation Flux Measurement (ECOR) system and the Energy Balance Bowen Ratio

(EBBR) system ((Sullivan et al., 1997); Ermold and Cook (1993)). Advection was quantified with observations from an array

of Doppler Lidar and AERI instruments at the ARM SGP site using the method outlined in Wagner et al. (2022). Radiative

heating was computed from TROPoe thermodynamic profiles using the rapid radiative transfer model RRTM (Mlawer et al.,

1997). W23 demonstrated that this approach to estimate entrainment agreed both with an observationally derived water vapor

entrainment flux derived from multiple ground-based lidars and to large eddy simulation (LES) output. However, entrainment

(as described above) is really composed of two terms: the flux of a property across the boundary separating the BL from the

free troposphere (ENT1) and the change in the concentration of a property as the depth of the BL changes (ENT2).

In a steady state BL, the depth is not changing, the ENT2 term is zero, and the residual can be used to estimate the entrainment

fluxes (ENT1). However, during the morning hours, the depth of the BL is usually changing relatively rapidly and thus ENT2

can no longer be assumed to be negligible. The current study aims to show that the residual assumption agrees well with the

sum of the two entrainment terms (ENT1 + ENT2) and highlight the importance of interpreting the residual of a MD as this

total entrainment rather than entrainment fluxes (ENT1) alone.

## 2   Methods

In this study, we apply a mixing diagram framework to the morning and afternoon BL and assess its closure and stochasticity.

To do this, we apply this method to large-eddy simulation output from single columns and from the entire horizontally averaged

(slab) output, which serves as the truth in this study. We also investigate how computing the mean temperature and humidity

only over the lower part of the mixed layer compares to when the entire mixed layer is used when computing the mean. Most of the analysis in this study focuses on 8 August 2017 case used by W23, which was synoptically quiescent and a part of the

Land Atmosphere Feedback Experiment (LAFE; Wulfmeyer et al. (2018)) that focused on land-atmosphere interactions. At the end of this study, results are considered for four additional dates during LAFE; i.e., 7, 14, 17, and 29 August 2017. The full analysis period is the time from just after the morning transition to just before the evening transition (0800 - 1700 CDT). This time period is split into morning hours (0800 - 1200 CDT) and afternoon hours (1200 - 1700 CDT), where the boundary layer depth is changing with time more rapidly during the morning hours than the afternoon hours.

## 2.1 Mixing Diagrams

In a mixing diagram (MD), the total evolution of the latent and sensible heats over a specified time period is plotted, and the components (large-scale advection and radiative tendencies, surface fluxes, and entrainment) are plotted as vectors on the same figure, allowing us to visualize the contributions of all the forcings to that diurnal evolution of sensible and latent heat. MDs are typically only used when the BL is convectively well-mixed. Closure is obtained when the sum of the components (i.e., on

the right-hand side of Eq. 1) equals the overall evolution in mean temperature and moisture within the well-mixed BL (i.e., the left-hand side of Eq. 1). We define "closure" as the length of the vector distance between the total evolution and the sum of the components, as can be seen by the red arrow in Figure 1. We can evaluate closure as the distance to close the sensible and latent heats individually or as a total vector value.

The mixed layer budget (Eq. 1) requires that the depth of the mixed layer (i.e., the depth of the BL, also denoted $z_i$) be well

defined and continuous. For this work, we compare three different definitions for the BL depth, which are shown in Figure 2: the level of neutral buoyancy of a surface-based parcel of air (green), the level where the maximum humidity variance occurs (orange), and the level where the minimum potential temperature flux exists (blue).

## 2.2 Large Eddy Simulations

This work uses single column output from large-eddy simulations to act as a proxy for ground-based observations. Informed by

DALES (Heus et al., 2010), UCLA-LES (Stevens et al., 2005), and PALM (Maronga et al., 2015), MicroHH (van Heerwaarden et al., 2017) is a high-resolution computational fluid dynamics simulation that supports both direct numerical simulations and large-eddy simulations (LES). These simulations are run on the graphical processing unit (GPU), enabling them to run much faster than other LES. MicroHH uses the anelastic approximation to solve the governing equations of conservation of mass, momentum, and energy (Bannon et al., 2006). LES is useful in testing observational hypotheses since it gives relevant variables



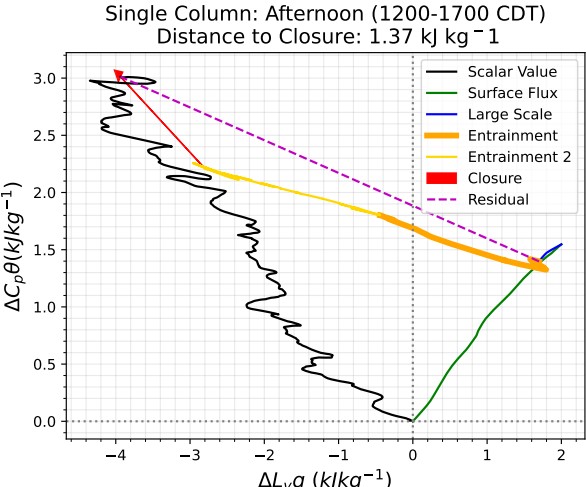

**Figure 1.** Mixing diagram showing the total evolution of the sensible and latent heats (black) along with the contributions to that evolution (surface fluxes in green, large-scale forcing in blue, entrainment 1 in orange, and entrainment 2 in gold), as derived from LES output where ENT1 and ENT2 are directly computed. The closure is the distance between the total evolution and the sum of the components (red arrow), and could be considered an error term in the total entrainment if it was computed as a residual (magenta).

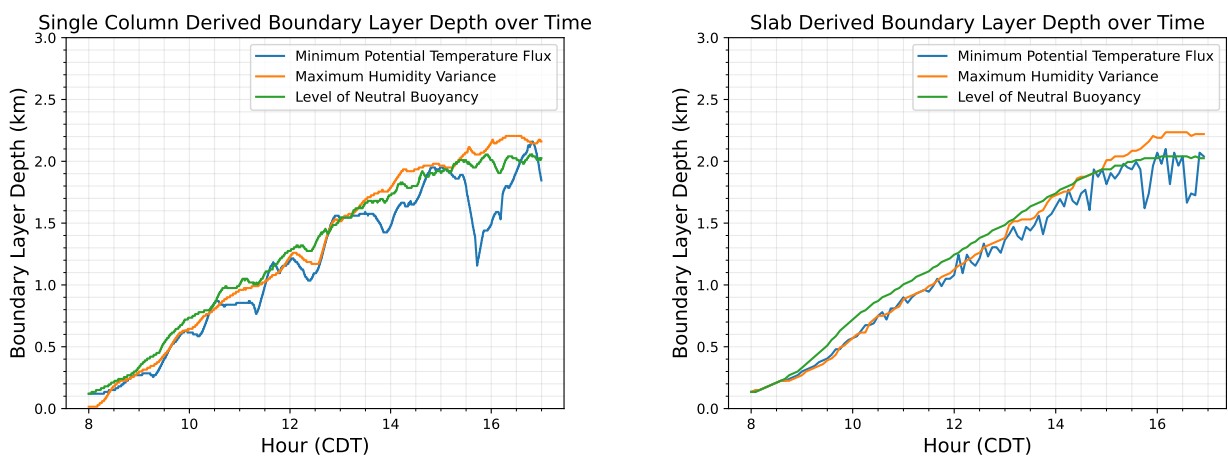

**Figure 2.** Comparison of three different BL depth definitions over the entire analysis period (0800 - 1700 CDT): the level of neutral buoyancy of a surface-based parcel of air (green), the maximum humidity variance (orange), and the minimum potential temperature flux (blue). Definitions are derived from the 30-minute temporal average of the instantaneous values of the single column (left) and from the 5-minute slab averages of values (right).

in all spatial and temporal dimensions. The data are internally consistent, and budgets that are calculated from these data must close by definition, down to the discretization error. In this study, we will use slab averaged values (i.e., averaged values over the entire LES domain at each vertical level) as the "truth". We want to investigate how real observations, which are making



time-series observations at a single point, are able to represent spatial statistics; thus, we are comparing statistics computed at a single location within the LES domain against the slab averages.

### 2.2.1 Model Configuration

MicroHH uses a second order central differencing spatial discretization scheme and a fourth order Runge-Kutta time integration method on an Arakawa C- grid. Potential temperature ($\Theta$) and water vapor mixing ratio ($q$) are carried as thermodynamic variables, and are conserved for adiabatic processes. The simulation uses ARM's constrained variational analysis (VARANAL) for initial and boundary conditions (Xie, 2004). VARANAL provides values for surface fluxes, large-scale advective and radiative tendencies that are spatially averaged over the entire ARM SGP domain. We set a 15 m vertical grid spacing from the surface to 4200 m, and 10 m horizontal grid spacing over a 6400 m domain size with periodic boundary conditions. The LES is run for 7, 8, 14, 17, and 29 August 2017 starting at 0300 UTC and run for 20 hours. Output from 64 individual columns that are equidistantly placed across the domain with 10 s temporal resolution act as profile observations.

### 2.3 Entrainment Fluxes

Since the LES columns do not yield both temperature and water vapor entrainment fluxes directly, we use the following for calculating ENT1 from the single column output:

$$\overline{w'\phi'}(t) = \sum_{t-t_{1/2}}^{t+t_{1/2}} \frac{1}{N} (\phi_{z=zi(t)} - \phi_{avg,z=zi(t)}) * (w_{z=zi(t)} - w_{avg,z=zi(t)}), \tag{2}$$

where $\phi_z$ and $\phi_{avg,z}$ represent a general property and the mean of that general property, while $w_z$ and $w_{avg,z}$ are the vertical velocity and the mean of the vertical velocity all at the top of the boundary layer. In this calculation, we estimate ENT1 using a time-series of data from a single column within the LES domain, choosing an analysis period averaged over a 1-h period centered on each 30-min to reduce the sampling uncertainty (Lenschow et al., 1994) Previous studies typically performed the time-series analysis along a constant height grid where $z_i$ was in the center of the temporal analysis window. Instead of taking the running average of a property at a constant height, we normalize the property by the BL depth first. This means that, while taking the running average of the property, the change in BL depth is already being considered. This method is outlined in more detail in Rosenberger et al. (2024).



## 3 Results

### 3.1 Residual Assumption

Figures 3 and 4 compare mixing diagrams for 8 August 2017 derived from three different BL depth definitions: the level of neutral buoyancy of a surface-based parcel of air (left), the maximum humidity variance (middle), and the minimum potential
temperature flux (right), for the slab output (top row) and the single column output (bottom row). During the morning time (0800 - 1200 CDT) shown in Figure 3, the magnitude of the total entrainment (ENT1 + ENT2) due to latent heat is approximately $8.5\ kJ\ kg^{-1}$ and for sensible heat it is $2.5\ kJ\ kg^{-1}$ regardless of the boundary layer depth definition used. There are, however, differences in the relative magnitudes of the ENT1 and ENT2 terms depending on the boundary layer depth definition. ENT1 is approximately $30\%$, $50\%$, and $60\%$ of the total entrainment (i.e., ENT1 + ENT2) when the level of neutral
buoyancy, minimum potential temperature flux, and the maximum humidity variance is used to define the top of the BL (i.e., $z_i$), respectively. To see the relationship with BL depth definition more clearly, Figure 5 shows the ratio of ENT1 to ENT1 + ENT2 for five different dates considered and for the three boundary layer depth definitions for the morning (left) and afternoon (right). Here, we see the relative contributions to ENT1 versus the total entrainment varies greatly with boundary layer depth definition. This tells us that the higher in the entrainment zone where $z_i$ is defined, the stronger the influence of ENT2.

We see similar results during the afternoon time (1200 - 1700 CDT) (Figure 4), where the overall magnitude of the total entrainment (ENT1 + ENT2) is the same regardless of boundary layer depth definition but the partitioning of the two terms changes with BL definition. During this time, the ENT2 term is largest using the maximum humidity variance definition. This is because, as we saw with the morning time, the ENT2 term is larger when $z_i$ is located higher in the entrainment zone, and at 1300 CDT the maximum humidity variance BL definition crosses the level of neutral buoyancy definition, making it higher in the atmosphere. The relative contribution of ENT1 to total entrainment remains the largest when the minimum potential
temperature flux definition is used, and the ENT2 term is the smallest in that case. Interestingly, the ENT2 is never negligible during either analysis period for this day. This is because the BL depth is still growing the entire time, so the ENT2 term must be considered.

Figure 5 also shows differences in the morning and the afternoon. During the morning, there is not much of a pattern across
the different dates of the different BL definitions, and the overall trend between the two ratios is mostly linear, but the ratios are not directly correlated. This is because the boundary layer is growing into the free troposphere. The evolution of the energy budget of $\theta$ and $q_t$ is happening at different rates. In the afternoon, however, the trend across all the dates and BL definitions is linear with very little deviation. This tells us that $\theta$ and $q_t$ are evolving at the same rate in the afternoon.





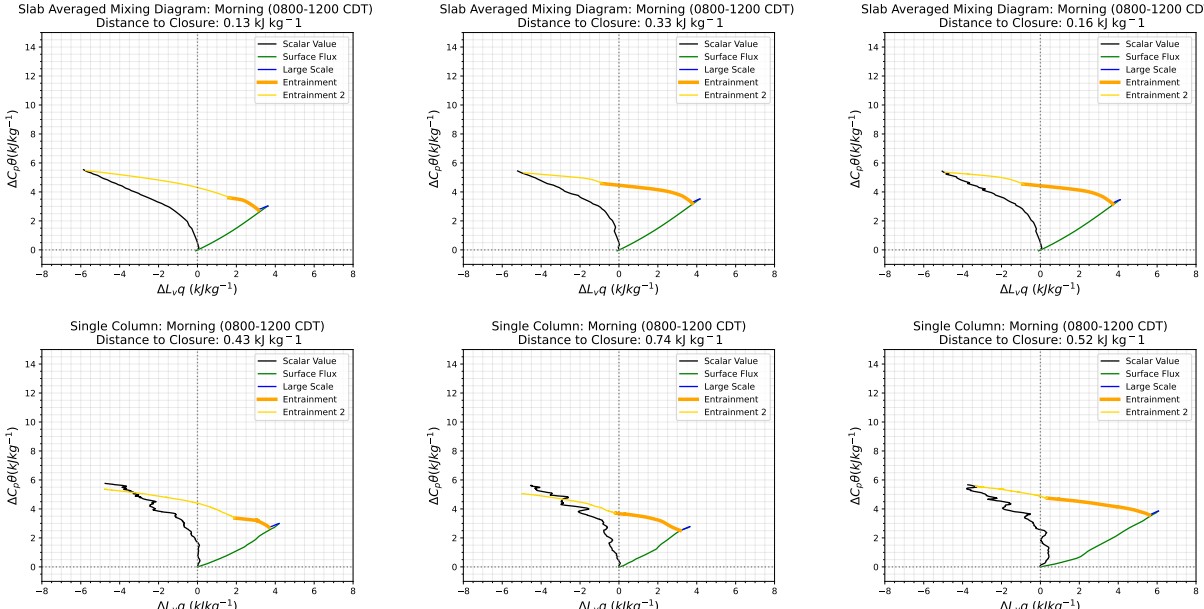

**Figure 3.** Mixing diagrams for the morning (0800 - 1200 CDT) of 8 August 2017 at SGP, comparing the impact of three different BL depth definitions: the level of neutral buoyancy of a surface-based parcel of air (left), the maximum humidity variance (middle), and the minimum potential temperature flux (right), for the slab output (top row) and the single column output (bottom row).

For both time periods, the single column MDs are slightly different from the slab averaged MD. It is difficult to capture the evolution of an entire horizontal domain from a single column, so differences between the two are to be expected. For both time periods, the partitioning between ENT1 and ENT2 is different for the maximum humidity flux and the minimum potential temperature flux cases in that the ENT1 term in the single column MD is larger than the slab for both of those cases. This tells us that defining $z_i$ as the level of neutral buoyancy, when calculated from that particular single column, is closer to the slab derived level of neutral buoyancy BL definition than the respective maximum humidity variance and minimum potential temperature flux definitions. This could be due to the fact that the variance and fluxes are averaged from single column output and then used to determine BL depth, while the level of neutral buoyancy definition uses direct single column output. The following section dives deeper into the variability across different columns and dates, and the level of neutral buoyancy definition is used in the remainder of this study.

### 3.2 Mixed Layer Definition

ENT2 represents the change in BL properties due to the change in the BL depth over time. The value of a property at the top of the BL is not the same as that of the property within the mixed layer. This difference is more drastic the greater the change in the





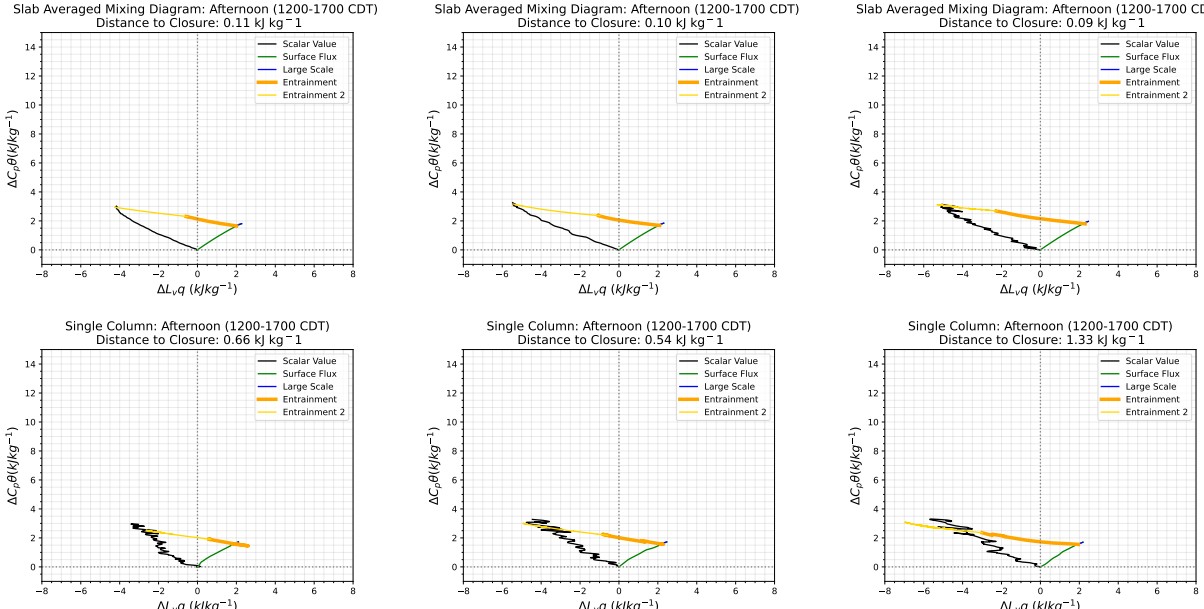

**Figure 4.** Same as Figure 3 but for the afternoon time period (1200 - 1700 CDT).

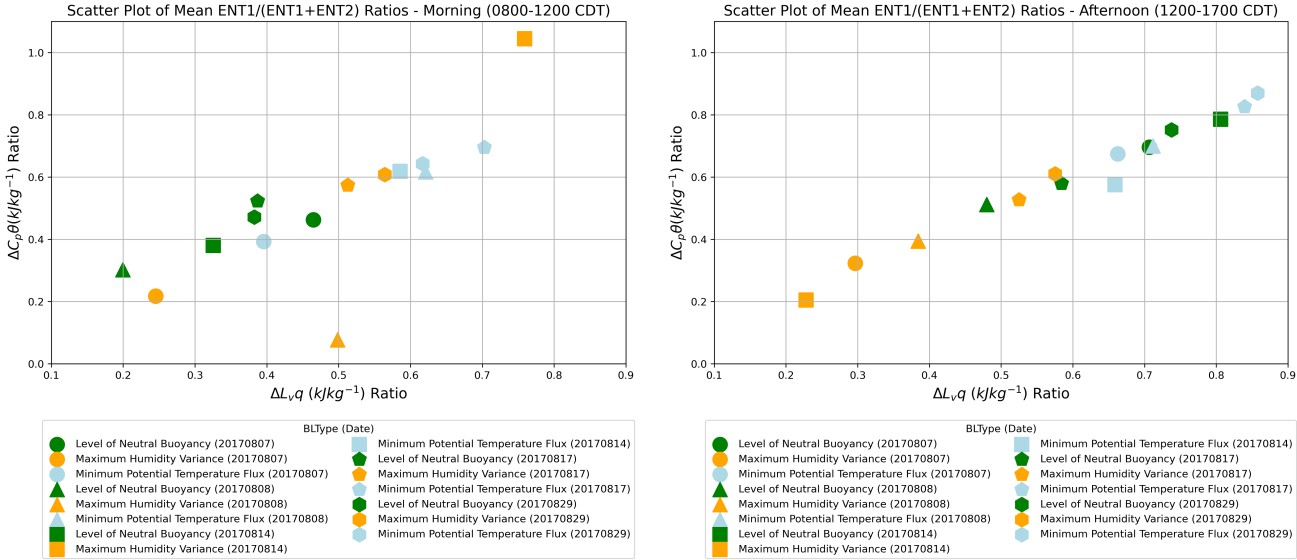

**Figure 5.** Ratio of ENT1 to (ENT1 + ENT2) for the morning (left) and afternoon (right) for $\Theta_t$ versus $q_t$. The different shapes are all five of the different dates considered, and the colors represent the three boundary layer depth definitions considered: the level of neutral buoyancy of a surface-based parcel of air (green), the maximum humidity variance (orange), and the minimum potential temperature flux (blue).

BL depth over time. ENT2 is crucial for closure in the mixing diagram, as shown in the previous section, and is calculated by $(\frac{\partial z_i}{\partial t} - \overline{w_i})\frac{(\overline{\phi}_{top} - \overline{\phi}_{ml})}{z_i}$. The difficulty here is defining the "mixed layer." The mixed layer could be considered as the entire area





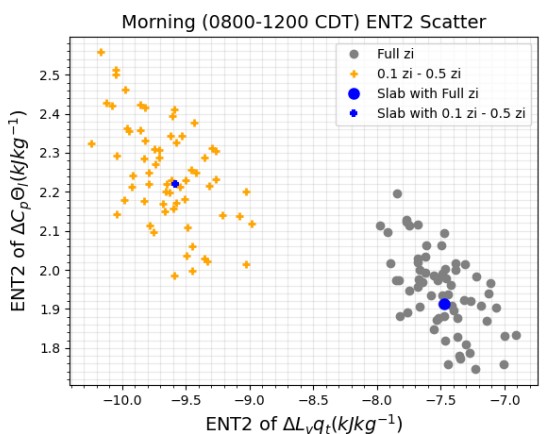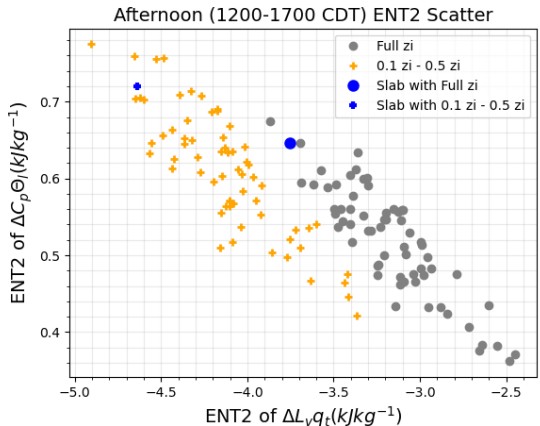

**Figure 6.** Comparison of the two different mixed layer definitions when calculating ENT2 for potential temperature and humidity for the morning (left) and afternoon (right) for 8 August 2017. The gray points are for when the mixed layer is taken as the entire area below $z_i$ (full $z_i$), and the orange $+$ is from when the mixed layer is defined as $0.1z_i - 0.5z_i$ (restricted) derived from each individual column in the domain. The respective slab ENT2 values are shown in blue.

below the top of the BL or, as in W23, it could be defined as $0.1z_i - 0.5z_i$. These will be referred to as the full $z_i$ and restricted

definitions respectively. Here we see which definition allows for a single column to better capture the slab ENT2. Figure 6

shows the second entrainment term for the morning (left) and afternoon (right) when the mixed layer is taken to mean the full

$z_i$ (gray) and when the mixed layer is restricted ($0.1z_i - 0.5z_i$, orange $+$) for each column, and compares them to the mean

second entrainment term of the slab average (blue ● and $+$ for the full $z_i$ and restricted cases, respectively). In the morning,

both definitions of the mixed layer mean for individual columns cluster around the slab value in both sensible and latent heats,

but the estimates of the ENT2 value is very different between the two methods. In the afternoon, there is much more overlap

between the two different methods, though that overlap tends to underestimate both ENT2 contributions. It makes sense that

the afternoon values would have more overlap between the two methods as the change in boundary layer depth over time is

smaller in the afternoon, making the magnitude of ENT2 smaller at that time, so the restricted mixed layer would have similar

values to the entire space below the BL during that time. In the remainder of this study, we use the full $z_i$ method regardless

of the time of day, as the cluster around the slab value in the morning is slightly tighter for this method than for the restricted

method, so single column values do a slightly better job of capturing the slab ENT2 with that definition.

## 3.3 Variability

No two columns will yield the exact same MD, due to sampling uncertainties. To get a sense of how well this MD framework

behaves for many individual columns, we compare the closure of the sensible and latent heat terms for a set of columns and



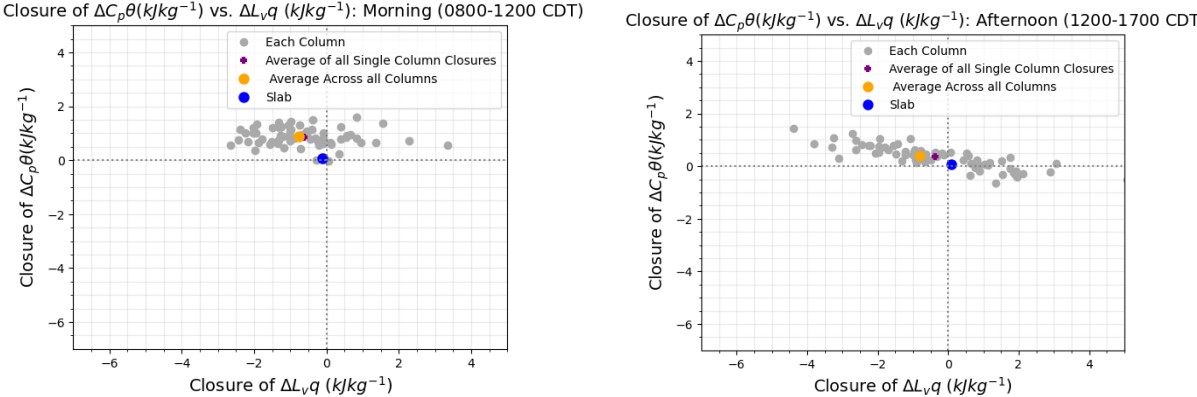

**Figure 7.** Scatter plots showing the closure in the sensible and latent heats for each single column (gray), the average of the closure values for all the columns (purple), the average across all 64 columns (orange), and the slab average (blue) for 8 August during the morning (left) (0800 - 1200 CDT) and afternoon (right) (1200 - 1700 CDT). In the morning, the average of all columns (purple) and the average across all 64 columns (orange) points are on top of one another.

various cases. We look at the closure of the slab averaged output, which is the LES slab averaged statistical output and serves

as the truth to which we compare all the other closure values. Then, within the LES domain, we have 64 individual columns

that serve as a proxy for observations. If we sum the data from those 64 columns and average over the horizontal domain, we

get the closest to the slab data that we can get from the individual columns, we call this the average across single columns.

Figure 7 shows the closure in the sensible and latent heats for the slab output (blue), from each individual column (gray), and

the average across single columns (orange). The purple dot is the average of all the closures of the individual columns (the

average of all the gray dots), and shows the general trend of all the columns. The left plot is for the morning (0800 - 1200 CDT)

and the right plot is afternoon (1200 - 1700 CDT) for 8 August. Ideally, our 64 individual columns would replicate the slab

average (our "truth") by temporally averaging single column output. We see that during the morning (left), the average across

the 64 columns and the average of all the columns is the same. This means that the average single column will yield a similar

closure value to the full array, but neither of these values perfectly aligns with the slab value. However, for this morning time,

both the orange and purple points show that these averages allow for nearly perfect closure in the latent heat. In the afternoon,

the average of all the columns is closer to the slab value than the average of all 64 columns, so for this case, the average single

column performs better than the full array. A positive closure value means the residual underestimates the entrainment; i.e.,

that the sum of the entrained heat and moisture is too small, while a negative closure value means the residual will overestimate

the entrainment; i.e., the sum of the entrained heat and moisture is too large. We see that in both the morning and the afternoon

time periods, we have more variability (i.e., that the spread of the individual columns relative to the slab averaged value) in

the latent heat than the sensible heat and that the average single column tends to underestimate the latent heat in relation to the





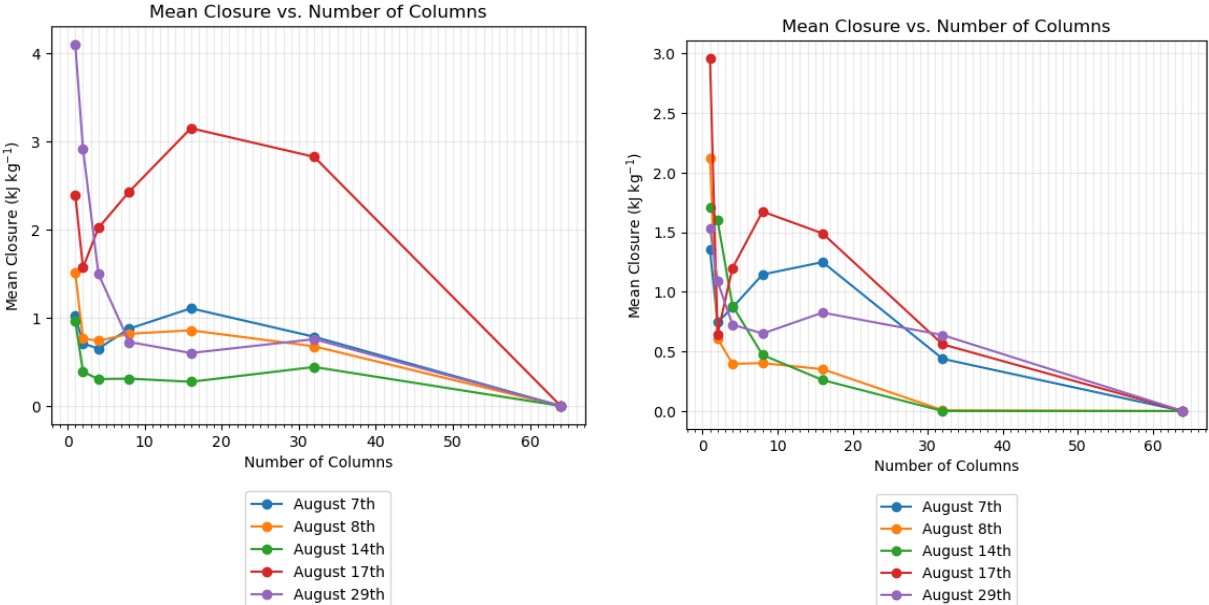

**Figure 8.** The mean closure value for a mixing diagram versus the number of columns used in calculating that closure for each of the five dates considered for the morning (left) and the afternoon (right).

slab average value of the latent heat. These figures give us a sense of the sampling error. The spread of the individual columns serves as an indicator of maximum uncertainty in the sensible and latent heats. Figure 8 shows the mean closure value versus

the number of columns used in calculating that closure across all five dates considered, over the entire day (0800 - 1700 CDT). For the most part, the more columns used, the smaller the closure value. This shows us that using multiple columns reduces the sampling error, or that using profilers, rather than one, would significantly reduce the sampling error.

To see the closure trends and compare them across different dates, Figure 9 compares the closure values for all five dates considered (7, 8, 14, 17, and 29 August 2017) for the morning (left) and the afternoon (right). The larger lines show the 1-$\sigma$

error bars on the single columns to better visualize the spread of the errors on each date. In the morning, the average across all the columns is closer to the slab values with respect to the latent heat but underestimates the sensible heat. In the afternoon, we see that the average across all the columns underestimates the sensible heat and tends to overestimate the latent heat across all the dates. This is consistent with what we saw in Figure 7, which confirms that our results are consistent across multiple dates.

Some reasons for the discrepancies between the slab value and the column values could be that there is a bias imposed by

temporally averaging the single column data, the method for extrapolating single column values to the surface, or the distance between the individual columns is large enough to not fully represent the domain.





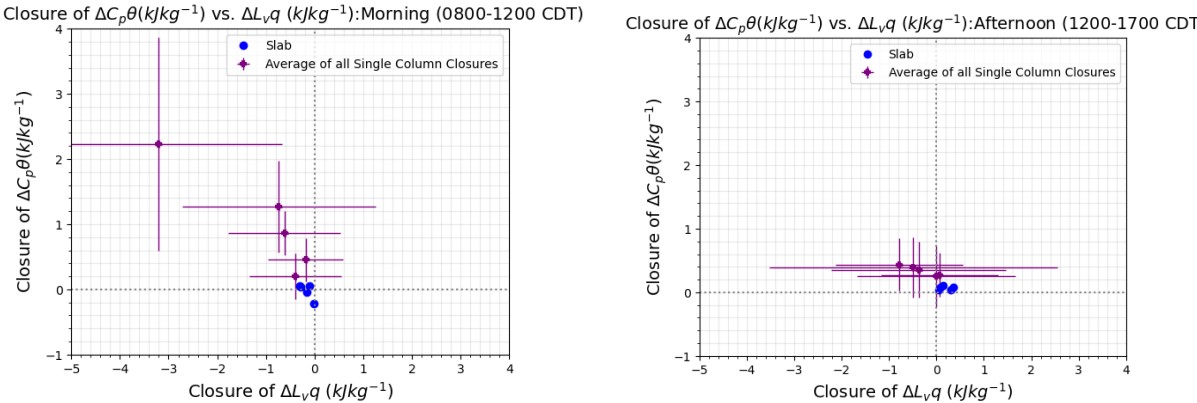

**Figure 9.** Comparison of the closure values averaged across all the columns of each of the five dates considered (7, 8, 14, 17, and 29 August 2017) (purple +) with a 1-$\sigma$ uncertainty range and the slab averaged closure values (blue ●) for the morning (left) and afternoon (right).

## 4  Discussion

Deriving entrainment using the MD framework as a residual assumption is valid during time periods where the BL depth is changing; however, that residual term does not represent only the entrainment fluxes (ENT1) but the total entrainment (ENT1 +

ENT2). ENT2 is crucial for closing a mixing diagram when the BL depth is changing. The magnitude of the total entrainment vector remains the same regardless of the BL depth definition being used, but the magnitudes of the individual terms change since different BL depth definitions sit in different positions within the entrainment zone. ENT2 is larger the higher in the entrainment zone the BL depth definition. Calculating ENT2 requires defining a mixed layer, and we found that using the full BL depth as the mixed layer definition yields results from a single column that are closer to that from the slab values than a

previously identified range of $0.1z_i - 0.5z_i$, assuming there are no systematic errors in the mean profiles of $\theta$ and q over the depth of the CBL (which is why W23 used the restricted height definition for that analysis).

MDs can be used to describe the evolution of the heat and moisture budgets of the BL where the BL depth is quasi-stationary or changing with time, and this method can be applied to single column output. The MDs derived from LES single column output tend to underestimate the amount of sensible heat and overestimate the amount of latent heat when compared to the slab

averaged LES output. The disagreement is important because the slab output is the truth, and the difference from that truth serves as a method for determining sampling error. Ultimately, we see that there is less sampling error for sensible than latent heat, and that there is more error in the latent heat in the afternoon than in the morning. The greater amount of sampling error in latent heat is expected because it is more sensitive to variability at $z_i$ than sensible heat. These results are consistent across multiple columns and multiple dates.

## 5 Conclusions

This work uses LES output as a testbed for determining entrainment from ground-based remote-sensing observations using a MD framework. By comparing results from proxy observations derived from LES single column output to the LES slab averaged output, we develop a method for applying a MD framework to morning hours and deriving total entrainment. We find that the residual from a MD framework represents the total entrainment and compares well with the actual sum of the entrainment terms ENT1 and ENT2. In future, it is crucial to interpret the residual of a MD as the sum of both entrainment terms rather than the entrainment fluxes alone.

The magnitudes of the ENT1 and ENT2 terms are sensitive to BL depth definition, but the magnitude of total entrainment stays the same regardless of definition. Using a mixed layer definition of the entire BL allowed for better agreement in calculating ENT2 from the single column and slab output. Finally, sampling error was estimated by determining the average closure value across multiple columns and across five individual dates, and we showed that the sampling error was larger for latent heat than sensible heat.

Finally, sampling error was reduced dramatically from a single column to multiple. Adding even a few more vertical profilers could drastically improve sampling error leading to more accurate and representative observations in the future. More work should be done to determine an optimum number and spacing for profilers to best reduce samling error.

*Code and data availability.* Code will be uploaded to a GitHub repository before the final review.

*Author contributions.* TH and DT conceived the concept, which was further advanced by all authors. TR performed the LES runs and analysis. TR wrote the manuscript draft, and all authors reviewed and edited the manuscript.

*Competing interests.* The authors declare that they have no conflict of interest.

*Acknowledgements.* This project was supported in part by the U.S. Department of Energy's (DOE) Atmospheric System Research (ASR), and Office of Science Biological and Environmental Research program, under (DE-SC0020114; DE-SC0024048) and the National Oceanic





and Atmospheric Administration's (NOAA) Global System Laboratory (89243019SSC000034) and Atmospheric Radiation Measurement (ARM) program, as well as by the NOAA Atmospheric Science for Renewable Energy Program. Data were obtained from the Atmospheric Radiation Measurement (ARM) User Facility, a U.S. Department of Energy (DOE) Office of Science user facility managed by the Office of Biological and Environmental Research. Computation was done at Cleveland State University's Center for Applied Data Analysis and

260    Modeling (ADAM).



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
