# Peer review of "An LES Exploration of the Assumptions used in Retrieving Entrainment from a Mixing Diagram Approach with Ground-Based Remote Sensors"

_EGUsphere, 2024_

## Author Comment (AC1)

The authors would like to thank the reviewer for their time in reviewing our work and for their thoughtful comments, questions, and recommendations. Our responses are in blue.

Reviewer 1 comments

This study investigates the assumptions of the mixing diagram (MD) approach when used to estimate entrainment (ENT) into the boundary layer (BL) from ground-based remote sensing by using LES single columns as proxies for these observations. The main assumption analyzed is that the residual of the MD can be used to represent the entrainment flux across the top of the BL. The authors show that for these simulations, the residual term not only represents the entrainment flux, called ENT1 in the text, but also a second term that takes into account the change in concentration of a given atmospheric property as the BL depth changes, called ENT2. The authors show that this second term, ENT2, is crucial in understanding the contribution of entrainment to mixed layer energy budgets, especially when the BL is in a growth phase (such as during the morning hours). Additionally, the authors analyze three separate approaches for calculating BL depth, and show that the using neutral buoyancy approach in the MDs leads to the most accurate estimation of the domain-mean ENT when just using output from a single column of the LES. Finally, the authors show that the estimation of domain-mean LES properties is improved when averaging over multiple columns when compared to only using a single column. I think the impact of this result could be highlighted more in the conclusions, as it shows that observations of spatial statistics could be much improved by just adding a few more ground-based observation sensors, as opposed to only using one, and could be useful for future observational studies.

After review, I recommend that this document be accepted after undergoing minor revisions. I think the science is looks good and is well-written. It constitutes a valuable addition to the journal, however, there are some points that need to be explained in more detail and with more justification, as well as some reorganizing that could be done to improve readability.

Minor comments

L35-41: There appears to be some inconsistencies in this equation and the definitions of each term that follows. There is a negative sign in front of the ENT1 flux term in eqn. (1), however, in the definition below (L41) this minus sign is absent, which is correct? Furthermore, in the last term in equation one, $<\varphi\_bar>$ is used to represent an averaged mixed layer property, but below, $\varphi\_ml$ is used. I would just pick one and be consistent.

Thank you for your attention to detail and for bringing our attention to these inconsistencies. We have corrected them.

L46-48: I think the end of this paragraph could benefit from a small explanation or example of what ENT2 means physically. ENT1 is explained fairly easily, as simply the flux of one property from the free troposphere into the BL. They explain it further down in L63 as a concentration change as the BL depth changes, but it might aid readability if this explanation was moved up here, or repeated here. I found myself not understanding what ENT2 was until I got to this explanation in L63.

Thank you for your comment. In line 47, we changed "ENT2 takes into consideration the change in BL properties due to the change in the BL depth over time." To "ENT2 accounts for the change in concentration of a property with a change in BL depth over time." We hope that this adds clarity to our description.

L83-84: The authors state that MDs are usually only used when the BL is well-mixed. In my opinion, this seems to muddle some of the later conclusions which highlight the importance of ENT2 in the morning growth phase. The BL is not always well-mixed, especially early in the morning when growth is fastest. I would add a caveat for this in the conclusions, or provide some justification that for these days, the BL is well-mixed during the analysis period if it is.

 Thank you for your comment and request for clarification. We believe that the BL is well-mixed for our given time period that begins an hour after sunrise and ends before sunset. We agree that the BL is not well-mixed during the early morning hours, and in future work, we would like to adjust our mixing diagram framework to extend to early morning and afternoon to evening transition time periods. The work done here makes a distinction between a quasi-stationary boundary layer and times where the BL is well-mixed, namely, the BL can be well-mixed while it is still growing. However, since MDs are showing the mean properties over a defined layer, a MD method can be applied to any layer – regardless of whether or not it is well mixed, as long as all terms are being considered. For example, since the BL is changing (namely growing) during our entire analysis period, ENT2 is crucial to obtain closure within our defined layer of surface to BL top. In highlighting the importance of ENT2 in closing the mixed layer budget, we have extended the time period over which a MD can be applied. We have changed the phrasing at lines 83-84 to say "MDs are typically only used when the BL is quasi-stationary, however this is very limiting as the BL is often changing over time. For applying a MD framework, the BL is considered well-mixed during time periods where the BL is steadily growing or decaying rather than rapidly growing and decaying as it does during morning and evening transition periods."

L97: I would remove the sentence here about the GPU and its effect on LES runtime. It doesn't seem relevant. I also think it could lead the reader to be skeptical and ask questions like: "If the LES runs so much faster than others, why didn't they do even more runs?"

Thank you for your comment. We understand the point and have removed the sentence.

L108: I know that the lower BCs are prescribed from VARANAL, but are they spatially homogeneous or heterogeneous (i.e. does each grid cell feel the same domain-mean fluxes)? I think this is key to the results, as spatial heterogeneity at the surface can significantly affect BL development over a domain of this size. If the domain is homogeneous, I would also add a caveat to the conclusions stating how they could change if spatial heterogeneity was included at the land-atmosphere interface.

Thank you for your comment. We have clarified that the surface fluxes from VARANAL are applied in a spatially homogeneous way at line 108. We have also added the following to the conclusions to reflect your recommendation. "These findings were all based on spatially homogeneous surface fluxes. Spatial heterogeneity would likely exacerbate our results of sampling error reduction with additional columns, as one column may be even less representative of an area in that case. Further work should be done in the future to determine the ways in which a heterogeneous surface would impact the overall evolution of sensible and latent heats throughout the day within the boundary layer."

L111: Why was 6400 m domain size chosen? This is a lot smaller than most climate model grid cells. Are you intending to help improve higher resolution models used for numerical weather prediction? Since the LES is used as a proxy over a spatial domain, what is the proxy meant to represent? Maybe I am reading into this too much, but I think you should justify your choices for domain size and resolution more with another sentence or two.

For this study, we believe it is more important to accurately simulate the atmospheric boundary layer and the entrainment zone, rather than mimic a climate model's grid column. We therefore chose to invest more in grid refinement than in domain broadening. With 6.4km being several times the boundary layer depth, we can still resolve the turbulence producing scales of the boundary layer, especially given that the diurnal cycle limits the growth of scales over time. We have added "To accurately simulate the diurnal evolution of the atmospheric boundary layer, we chose a 6400 m domain size that is several times the BL depth but still able to resolve turbulence producing scales of the BL (Fedorovich et al. 2004)."

L139: The term "entrainment zone" is used without being defined, I have an idea of what it means, but any reader might not be entirely sure without a solid definition. I would add a sentence to do that.

Thank you for your comment. We agree that offering a definition of the entrainment zone would be helpful here. We've added "The entrainment zone is a layer of intermittent turbulence between the mixed layer and free troposphere where the potential temperature gradient is strongest and where the buoyancy flux is negative (Figure 3, see Stull 1988). The variation in ENT1 and ENT2 magnitudes tells us that the higher in the entrainment zone $z_i$ is defined, the stronger the influence of ENT2." on line 139. We have also added the following figure to show the entrainment zone.

[Figure]

Figure 5: This figure is a little confusing/overwhelming at first glance, and the key doesn't help much. This is just a suggestion, but I think it would be easier to understand if the days were the variable that was color-coded, and the type of BL depth method used was represented by the different shapes (the reverse of what is used now). That way, you could have a much simpler key which just stated which day was what color, and which method was one of the three chosen shapes, I don't think you need to have a line for each symbol/color combination in the key. This would get rid of the long key used presently, which has many repeated terms and in my opinion only confuses the reader more initially.

Thank you for the excellent suggestion to make this figure easier to read. We have implemented your suggestions.

L179-181: You say that the cluster around the full z_i method is "tighter", however its hard for me to conclude this based on the figure alone. It might just look tighter because the dots you use for the full method are larger than the "+" signs used for the restricted. I think finding a way to quantify this clustering would strengthen this argument, and help in justifying choosing the full method over the restricted method.

Thank you for your point. We agree that we needed to quantify the sizes of the clusters. We found this by calculating the pairwise distance across each cluster and included those distances in the text, starting at line 180. "For the morning, the pairwise distance of the respective clusters is calculated, and for the restricted method that distance is 1.28 ($kJkg^{-1}$), and for the full method, that distance is 1.10 ($kJkg^{-1}$). In the afternoon, there is much more overlap between the two different methods, though that overlap tends to underestimate both ENT2 contributions. The pairwise distance for the restricted method in the afternoon is 1.58 ($kJkg^{-1}$) and for the full method, it is 1.45 ($kJkg^{-1}$)." We changed "tighter" to "as the cluster around the slab value in the morning is closer, according to the pairwise distance of farthest points in the cluster" in line 187.

L194: This is nitpicky, but you say that the two averages are "the same", however, in the figure they look very close, but not *the same*. I would change this wording to reflect this.

Thank you for your very valid comment. We have adjusted the phrasing to say that the points are very close rather than the same.

L195-199: I don't really understand the point the authors are trying to make here. How do the orange and purple points show that there is "perfect closure" for the latent heat flux in the morning? And the next sentence, how do they show that the "the average single column performs better than the full array"? I think reasoning needs to be explained more or maybe reworded, because as it is right now these two sentences confused me.

We apologize for the confusion. You are correct in pointing out that neither point perfectly aligns with the latent heat value of the slab results – we have removed that sentence. What we are trying to show is whether a randomly placed profiler (average of single column) would more accurately represent a given area than averaging the results across many profilers (average across all columns). We have changed the phrasing to make this clearer. It now reads, "We see that during the morning (left), the average across the 64 columns and the average of all the columns is very close. This means that the average single column will yield a similar closure value to the full array. In the afternoon, the average of all the columns (purple) is closer to the slab value (blue) than the average across all 64 columns (orange), so for this case, the average

single column replicates the slab values better than the result from averaging across the full array."

L207: This is just for clarity, I would add in the word "multiple" before profilers. So the sentence would read "...using multiple profilers, rather than one..."

Thank you for your suggestion. We have added "multiple" to that sentence to make it more clear.

Figure 9: Why not differentiate day by color or symbol here? Like in Figure 5.

Thank you for your recommendation. We have done this and think it makes the figure more helpful.

L233: "The greater amount of sampling error in latent heat is expected because it is more sensitive to the variability at $z_i$..." I think this needs more explanation. LH is more sensitive to the variability of what? And why is this expected? Why is the LH more sensitive to this variability? Is there a previous study you could cite for this?

Thank you for your questions. We have changed "variability" to "mixing with the free atmosphere" and hope this makes our point clearer.

L247-249: I think you could highlight this conclusion a little more, and go into more detail on its impact on future observational studies.

Thank you for your comment. We have gone into more detail in describing the implications of this work for future observational studies. "Sampling error was reduced dramatically from a single column to multiple. This shows us that being able to average over more than one single column would be more representative of a selected region. The implications of this result for observations is that adding even a few more vertical profilers to a region could drastically reduce sampling error. This would lead to more accurate and representative observations in the future. Future modeling work should be done to determine an optimum number and spacing for profilers to best reduce sampling error."

---

## Author Comment (AC2)

The authors would like to thank the reviewer for their time in reviewing our work and for their thoughtful comments, questions, and recommendations. Our responses are in blue.

Reviewer 2 comments

The article is motivated by the need to accurately model the boundary layer entrainment flux and changes in the property concentration due to entrainment in the budget of scalar property equations for use in numerical weather prediction models.

The authors aim to robustly estimate the entrainment fluxes from ground based remote sensing instruments. To this effect, the validity of the residual assumption used by Wakefield et al 2023 (W23) is investigated over the course of the evolving BL using LES framework, and a reinterpretation of the mixing diagram method presented in W23 is offered.

The scientific method, the results, and the discussions that follow are compelling. However, the article suffers from a few minor shortcomings that hamper readability. I recommend that the authors carefully consider below listed minor suggestions/revisions in their resubmission to the journal:

- Line 53-55: "Because of the vertical resolution of the profiling instruments used in W23, they computed the mean of the mixed layer properties only from 0.1zi−0.5zi, where zi is the depth 55 of the BL determined from the TROPoe retrievals using a parcel method". Please clarify what is the instrument vertical resolution. Is this superior to the LES resolution? Do the authors wish to suggest that the data resolution used by W23 is insufficient to resolve the BL? I fail to understand the point of this statement.

Thank you for highlighting this section. We apologize for the confusion here and have corrected our statement to properly reflect the work that was done in W23. The sentence now reads "To avoid the surface layer and entrainment zone, W23 computed the mean of the mixed layer properties only from $0.1 z_i - 0.5 z_i$, where $z_i$ is the depth of the BL determined from the TROPoe retrievals using a parcel method."

- Line 67: MD here is used without defining the abbreviation. The abbreviation is defined in section 2.1. Please move the definition of MD to line 67 instead for clarity.

Thank you for your close reading. We have added the definition to line 67.

- The end of section 1 could very much use a breakdown of the article sections to guide the reader through the document.

Thank you for your suggestion. We have added "Section 2 describes the methods used in this study. Section 3 presents the results of whether the residual assumption for deriving entrainment is valid, a comparison of different definitions of the mixed layer for calculating ENT2, and variability across different dates and boundary layer depth definitions. Section 4 offers a discussion of the results. Section 5 highlights conclusions and presents opportunities for future work on this topic. " to the end of section 1.

- A nearly periodic drop in the BL depth is observed when minimum potential temperature flux method is considered for single column (left plot of figure 2). See at times 0930, 1030, 1130, 1230, and 1400 hour time stamps. Is there an obvious explanation for such an observation? Such a systematic trend could potentially affect the relative magnitudes of ENT1 and ENT2 for single column data and brings into question the representativeness of the BL depth estimate using the minimum potential temperature flux method. (Overall, section 2 is well designed to crisply convey the investigation methodology).

Thank you for your observation. Since the single column fluxes are being calculated over a 30-minute window size, this could be the result of a rounding error. In Rosenberger et al. 2024, they used this method of calculating higher order moments from single column output and found that the fluxes had more variability than higher order moments such as variance and skewness.

- Are ENT1 and ENT2 estimates presented in Figure 5 computed from single-column data, slab averaged data, or average of the equidistantly placed columns? Please specify this upfront when describing figure 5. (If such a detail was already resented in section 2 earlier, ignore this comment).

Thank you for your comment. These ratios come from the slab averaged data, and to reflect this, we have changed the sentence on line 136 to read "Figure 5 shows the ratio of the slab derived ENT1 to ENT1+ENT2…"

- Further, in figure 5: There appears to be no systematic trend in the ENT1/(ENT1+ENT2) ratios using the three methods even in the afternoon data when compared across the 5 different days. For instance: for days represented by the square and circles, the BL estimated using the maximum humidity variance seems to provide consistently the smallest ratio of ENT1/(ENT1+ENT2) while the BL depth estimated using level of neutral buoyancy provides the largest ratios. Is there a specific reason as to why such a trend is not observed for the other 3 days?

Thank you for your question. We found that the ratio between ENT1 and the total entrainment depends on where the BL depth falls in the entrainment zone, so when one BL depth is higher than another, it will have a smaller contribution from the ENT1 term. The level of neutral buoyancy is not always going to be the deepest definition. We are not arguing that, in using a specific definition, there will always be a larger contribution to the ENT1 term, rather, we see that the contribution is dependent on the relative depths of the BL definitions.

- Lines 179-181 are vague. "Slightly better job" is purely qualitative here. It is hardly distinguishable from visual inspection of figure 6, especially in the afternoon data. Some effort to quantify such a difference would be more defensible and provide further evidence to the use of full $z_i$ as a proxy for the mixed layer.

Thank you for your comment. We agree that we need to quantify the distance of the respective clusters in Figure 6. We found this by calculating the pairwise distance across each cluster and included those distances in the text, starting at line 180. "For the morning, the pairwise distance of the respective clusters is calculated, and for the restricted method that distance is 1.28 (kJkg-1), and for the full method, that distance is 1.10 (kJkg-1). In the afternoon, there is much more overlap between the two different methods, though that overlap tends to underestimate both ENT2 contributions. The pairwise distance for the restricted method in the afternoon is 1.58 (kJkg-1) and for the full method, it is 1.45 (kJkg-1)." We changed "tighter" to "as the cluster around the slab value in the morning is closer, according to the pairwise distance of farthest points in the cluster" in line 187.

- I fully concur with RC1's comments that lines 193-196 are confusing and need a revision. This is only clear after reading it numerous times, but not readily intuitive from visual inspection. A brief explanation here is warranted.

Thank you for seconding the RC1's comment on this section. We have changed the phrasing here to say, "We see that during the morning (left), the average across the 64 columns and the average of all the columns is very close. This means that the average single column will yield a similar closure value to the full array. In the afternoon, the average of all the columns (purple) is closer to the slab value (blue) than the average across all 64 columns (orange), so for this case, the average single column replicates the slab values better than the result from averaging across the full array."

- The mean closures presented for August 17th dataset appears as a clear outlier. Is there a specific reason for this?

Thank you for your question. The 17th has a rapidly changing latent heat surface flux while the boundary layer is shallower than that of the other dates. We believe that this is due to the fact that there was rainfall at night on August 16. There was no rainfall on any of the other dates preceding the date that was selected. By adding additional columns into the mean, the degree of closure improves. The reason that it increases at first is likely due to the specific columns selected in the calculation.

[Figure]

[Figure]

[Figure]

[Figure]

- In figure 9 it is not possible to differentiate between the different days of the dataset. It would be nice to differentiate the legends of the closure estimates for average of single columns for clarity.

Thank you for your suggestion. The points in figure 9 have been color coordinated by date.

- Lines 221-223: "ENT2 is larger the higher in the entrainment zone the BL depth definition." This sentence doesn't make sense to me. What do the authors intend to say here?

Thank you for your comment. We see that, when the boundary layer depth definition falls higher in the entrainment zone, the magnitude of the second entrainment term is larger than the first entrainment term – or that the component of total entrainment due to ENT2 increases closer to the free troposphere. To make this point clearer, we have added "The magnitude of ENT2 is larger the higher in the entrainment zone the BL depth definition. This means that the contribution to the total entrainment from ENT2 increases the closer to the free troposphere the BL depth definition."

I recommend that the article be published subject to the authors addressing the above mentioned minor suggestions/revisions.